# Recent Advances in Extracellular Vesicle-Based Therapies Using Induced Pluripotent Stem Cell-Derived Mesenchymal Stromal Cells

**DOI:** 10.3390/biomedicines10092281

**Published:** 2022-09-14

**Authors:** Giuliana Minani Bertolino, Marie Maumus, Christian Jorgensen, Danièle Noël

**Affiliations:** 1IRMB, University of Montpellier, INSERM, 34295 Montpellier, France; 2Bauerfeind France, IRMB, 34295 Montpellier, France; 3Clinical Immunology and Osteoarticular Disease Therapeutic Unit, Department of Rheumatology, CHU Montpellier, 34095 Montpellier, France

**Keywords:** regenerative medicine, manufacturing, induced pluripotent stem cells, mesenchymal stromal cells, extracellular vesicles

## Abstract

Extracellular vesicles (EVs) are being widely investigated as acellular therapeutics in regenerative medicine applications. EVs isolated from mesenchymal stromal cells (MSCs) are by far the most frequently used in preclinical models for diverse therapeutic applications, including inflammatory, degenerative, or acute diseases. Although they represent promising tools as cell-free therapeutic agents, one limitation to their use is related to the batch-to-batch unreliability that may arise from the heterogeneity between MSC donors. Isolating EVs from MSCs derived from induced pluripotent stem cells (iMSCs) might allow unlimited access to cells with a more stable phenotype and function. In the present review, we first present the latest findings regarding the functional aspects of EVs isolated from iMSCs and their interest in regenerative medicine for the treatment of various diseases. We will then discuss future directions for their translation to clinics with good manufacturing practice implementation.

## 1. Introduction

Extracellular vesicles (EVs) are being largely investigated with regard to their capacity to carry a large number of molecules (proteins, lipids, different RNA species, and DNA) that play a role in intercellular communication and might be of interest for therapeutic purposes [1]. Thanks to their presence in biological fluids, they might be used as biomarkers of diseases; they may also serve as biological targets to decipher pathophysiological processes or as therapeutic agents. In the latter situation, EVs derived from mesenchymal stromal cells (MSCs) are likely the most investigated type of EVs for diverse therapeutic applications, including inflammatory, degenerative, or acute diseases (for review, see [2]). While the number of clinical trials with MSC-based therapies has rapidly risen for a variety of indications, few of them have reached phase III. Only three phase III trials on graft-versus-host disease (GvHD) and Crohn’s disease have been completed and published so far [3,4]. Variability in clinical outcomes was attributed to variability in the functionality of MSCs related to the tissue source, donor, or manufacturing process. It can be anticipated that such variability will also be encountered when using MSC-derived EVs (MSC-EVs), which are being evaluated in some clinical trials [5].

This raised the need for reducing the heterogeneity between MSC donors during ex vivo expansion and maintaining the critical quality attributes (CQAs). Selection of MSC donors or MSC subpopulations, as well as enhancing MSC functions by diverse techniques (priming, genetic modifications, etc.) could be an option (for review, see [6]). Another option might be to use a universal donor source, such as induced pluripotent stem cells (iPSCs) with theoretically unlimited access (Figure 1). iPSCs can generate MSCs in vitro and, genetic engineering can create iPSC lines with the capacity to evade the immune system as discussed in [7]. The use of induced pluripotent stem cell-derived MSCs (iMSCs) to produce EVs might even not require the generation of immune-privileged iPSC lines since EVs have been reported to some extent to be low or not immunogenic [8]. In this review, we will focus on the latest findings regarding the functional aspects of EVs isolated from iMSCs and their interest in translational medicine.

## 2. The Interest in EVs for Therapeutic Applications

EVs are divided into three main subpopulations: exosomes that are small-sized (50–150 nm) and contained in the endosomal compartment within multivesicular bodies that fuse with the cell membrane for EV release; microvesicles or microparticles that are medium-sized (100–1000 nm) and are produced by outward budding of plasma membrane; and apoptotic bodies that are the largest vesicles (100–5000 nm) coming from the blubbing of apoptotic cells [9]. EVs are released by all types of cells and present in all biological fluids where they were first considered waste products. They are now recognized as major actors in the intercellular communication pathways through the transport of many types of molecules that can be taken up by target cells in the microenvironment and influence cell function. Thanks to the specific content of their cargoes, they reproduce the main functions of parental cells and can be exploited as delivery agents of endogenous therapeutic molecules [10].

EVs present several advantages over the direct use of parental cells. Since they do not self-replicate, there is no risk of tumor formation after administration nor of ectopic differentiation. Due to their small size, they can be easily biodistributed and can cross the blood–brain barrier [11]. Similar to liposomes, they can be genetically modified to express ligands at their surface for receptor-mediated tissue targeting or to deliver therapeutic molecules, such as proteins, mRNAs, or miRNAs [12]. However, contrary to liposome-based drug delivery, one major interest of EVs is their capacity to be captured by the target cells where they can efficiently deliver their cargo since they possess the machinery to escape the degradation pathways in the lysosomal compartment [13]. In addition to their endogenous capacity to deliver functional cargo, they have been shown to be well-tolerated upon administration, even when repeatedly injected in mice [14,15]. They can also be long-term stored without losing their bioactivity even after freeze-drying, which makes them very convenient for off-the-shelf use [16]. Finally, EVs can be obtained from immortalized cells without the loss of their functional properties and the risk of promoting tumor growth [17].

Various sources of EVs from different cell types (dendritic cells (DC), tumor cells, etc.) have been tested in preclinical settings. DC-derived EVs have been shown to elicit an immune response from the recipient and may act as a vaccine [18]. By contrast, MSC-EVs display immunomodulatory and regenerative functions [19]. In fact, MSCs from diverse origins, including adipose tissue, umbilical cord, and bone marrow, are the major sources of cells for producing EVs with therapeutic potential. Several clinical trials are currently evaluating the safety and efficacy of MSC-EVs in regenerative medicine (reviewed in [5]). Of note, thirteen out of thirty-three registered trials are evaluating the use of MSC-EVs for COVID-19 or Acute Respiratory Distress Syndrome treatment (Table 1). Despite the establishment of selection criteria aiming to guide the isolation and expansion of MSCs, it is now a consensus that there is substantial heterogeneity between donors, evidenced by transcriptomic, proteomic, and epigenomic studies that likely induce variability in EV batches [20]. The main causes for such variability are the distinct tissue of origin and individual differences regarding genetic background, age, and health. This heterogeneity might be overcome by the use of iMSCs. Together with the strategies to lower the heterogeneity between MSC sources, stringent quality controls for standardization of the process will assume better reproducibility between batches (Figure 1).

## 3. iMSCs for Regenerative Therapies

### 3.1. Interest in Using iMSCs vs. MSCs

iPSCs were firstly generated in 2006 by introducing four specific genes (Myc, Oct3/4, Sox2, and Klf4, now collectively named Yamanaka’s factors) [21]. The four factors were reported to convert somatic cells into pluripotent cells that are functionally and morphologically analogous to embryonic stem cells and can differentiate into all three germ layers. Since their breakthrough, iPSCs have been extensively studied as cell sources for various applications, such as tissue graft, blood transfusion, and cancer immunotherapies. The growing interest in iPSCs comes from the fact that they have stemness features and therefore can be expanded without losing their pluripotency. They can be easily genetically manipulated with high efficiency and high reliability, giving the feasibility to generate a limitless supply of iPSCs from a single blood or tissue donor for the large-scale generation of standardized derivates [22].

MSCs originating from a single iPSC clone are more homogeneous than tissue-derived primary MSCs, and their biological functions are more predictable since their molecular signature is more stable in batch-to-batch comparisons [23,24]. Moreover, a large number of iMSCs can be obtained by exploiting the expansion potential of iPSCs before differentiation into MSCs, which enables the production of large, clinical-grade quantities of iMSCs without overly expanding the MSCs themselves. Hence, commercial amounts of minimally expanded MSCs from a single iPSC line have been revealed approachable in a routine manner, encouraging the use of iMSCs in clinical trials for diverse applications in regenerative medicine [24]. iMSCs display some functional differences with tissue-derived MSCs, in particular decreased propensity for adipogenic differentiation [25,26,27,28]. The gene expression profiles of primary MSCs and iMSCs are very similar, but iMSCs show reduced capacity in suppressing T cell proliferation, suggesting a lower immunomodulatory property [27]. Nevertheless, age-related DNA methylation that was accumulated during in vitro culture was overall reset in iMSCs [29,30], consistent with a stronger regeneration ability of iMSCs in animal disease models and higher proliferation rates when compared to primary MSCs [31,32,33,34].

To evaluate the senescence-associated phenotype of iMSCs, a recent study compared culture expansion of primary MSCs and iMSCs, reprogrammed from the same MSC donors until they reached growth arrest. With increased numbers of cell passages, both cell types acquired similar changes in morphology, in vitro differentiation potential, senescence-associated β-galactosidase activity, DNA methylation status, overlapping gene expression, epigenetic signature, and metabolic changes, particularly in senescence-related pathways [32]. The data suggest that iMSCs undergo senescence-related changes as do primary MSCs, while cellular aging may be controlled in iMSCs by expanding iPSCs in the pluripotent state to then generate relevant cell numbers of iMSCs of low passage. That means that iMSCs can be a mass-produced product with high batch-to-batch consistency that can result in robust clinical outcomes. Nevertheless, there are several methodological options for producing iMSCs, including MSC switch, embryoid bodies, and specific differentiation, but no unique consensus method has been selected for the scale-up production of iMSCs compliant with Good Manufacturing Procedures (GMPs) [35]. Therefore, the development of an optimized and monitored process for the generation of iMSCs can provide great opportunities for improving our knowledge about MSC biology, as well as widening the feasibility and application of MSC technologies. iMSCs represent a potential for the generation of MSC-based “off-the-shelf” products under GMP procedures and guaranteed CQAs, which are essential in therapeutic applications.

### 3.2. Preclinical Data Using iMSCs

The functional properties of iMSCs have been compared to tissue-derived MSCs, in particular their immunomodulatory capacity. Early studies have shown that iMSCs can inhibit NK cell proliferation and cytolytic function in a similar fashion as bone marrow-derived MSCs (BM-MSCs) while being more resistant to NK cell-mediated killing [36]. iMSCs and BM-MSCs exerted similar immunomodulatory effects on DC differentiation and function, but the expression of the maturation markers CD86 and CD40 were more decreased by BM-MSCs, meaning that different mechanisms could be expected relative to the cell origin [37]. After licensing by the inflammatory milieu, iMSCs could upregulate programmed death-ligand 1 (PD-L1) and indoleamine 2.3-dioxygenase (IDO) as a response to exposure to interferon-γ (IFNγ). As a result, they dampened the proliferation of peripheral blood mononuclear cells (PBMCs) and the expression of activation and differentiation markers, supporting the idea that iMSCs can exert functional immunoregulatory actions in vitro [24]. However, compared to BM-MSCs, human iMSCs were less sensitive to IFNγ-induced human leucocyte antigen (HLA)-II expression, and the analysis of cell retention and inflammation suggested that iMSCs had lower immunogenicity after in vivo transplantation when compared to BM-MSCs [38].

Similar effects of iMSCs and tissue-derived MSCs were also confirmed in vivo in several disease models. The administration of single or dual doses of iMSCs significantly attenuated disease severity and enhanced survival in a GvHD) mouse model [24]. Transplanting iMSCs into mice significantly attenuated severe hind-limb ischemia and promoted vascular and muscle regeneration, by increasing myogenesis and neovascularization [39]. The benefits of iMSCs were superior to those of BM-MSCs on limb ischemia in mice that displayed significantly less muscle fibrosis, inflammation, and more pronounced muscle regeneration. iMSCs also had the capacity, when implanted into surgically created periodontal defects in rats, to increase the amount of regeneration and newly formed mineralized tissue [40]. iMSCs promoted mucosal healing in colitic mice via the secretion of TNFα-stimulated gene-6 (TSG-6), one of the main candidates for the beneficial effects of MSCs [41,42]. A comparative analysis between urinary epithelial cell-derived iMSCs and umbilical cord (UC)-derived MSCs has further shown that iMSCs had a superior migration capacity in in vitro assays [33]. However, an evaluation of iMSCs and BM-MSCs from the same donor showed that iMSCs display a unique expression pattern of mesenchymal and pluripotency genes, with downregulation of those involved in trilineage differentiation [26]. BM-MSCs generally outperform iMSCs on trilineage differentiation in vitro [25,26,27,28]. Several studies comparing MSCs from multiple tissues and donor ages have found that iMSCs have overlapping developmental processes with young MSCs of fetal or UC origin, acquiring a rejuvenation gene signature by the loss of age- and tissue-related DNA methylation profiles [27,29,30,39]. These data suggest that iMSCs may not be the best candidates for applications aiming at direct cell differentiation, but they exert a beneficial regenerative effect in different inflammatory settings, configuring these cells as a potent tool for clinical applications. The use of iMSCs in clinical studies may reduce the impact of heterogeneity caused by age- and tissue-related epigenetic and microenvironmental patterns, providing more standardized cell products and overcoming possible issues from the clinical application of primary MSCs.

### 3.3. Perspectives toward Clinical Applications of iMSCs

From the perspective of clinical use of iMSCs, some safety concerns have been raised. Regarding invasiveness and ethical issues of tissue sampling, iPSCs are easily obtained from human samples, such as urine, and seem to be limitlessly expanded even from the small availability of starting material [23,43]. Safety concerns related to the generation of iPSCs using a retrovirus-based approach have now been addressed with the development of reprogramming protocols based on nonviral gene delivery systems [44,45]. A further key concern when using a pluripotent source for cell therapy is the potential for teratoma formation, but, for the moment, no study has reported tumor formation from iMSCs. In fact, during MSC differentiation, iPSCs lose the expression of pluripotent markers, such as Oct4, Nanog, and Sox2 [39].

Up to now, there are two registered clinical trials testing iMSCs, both designed by Cynata Therapeutics and using iMSCs produced by the patented Cymerus™ platform technology. MSCs are generated from intermediate allogeneic cells, mesenchymoangioblasts, which are derived from iPSCs. The iPSCs are derived from blood and were reprogrammed using a transgene-free, viral-free, and feeder-free technique [24]. The first clinical trial investigating the safety, tolerability, and efficacy of iMSCs (phase I, no. NCT02923375) in treating steroid-resistant acute GvHD in adults was finalized and had promising preliminary results [23]. No serious adverse event related to cell administration was observed, and 53.3% of the patients showed a complete response at day 100. In the second clinical trial, the company is recruiting patients to study the early effects of the intravenous administration of iMSCs in adults admitted to an intensive care unit with respiratory failure due to COVID-19 or another underlying cause (phases I/II, no. NCT04537351). A third clinical trial was recently announced, but not yet registered, sponsored by the University of Sydney using Cynata’s iMSCs, which will evaluate the effects of iMSCs in osteoarthritis in a phase III study (https://files.cynata.com/629/201111Phase3OsteoarthritisClinicalTrialCommences.pdf) (accessed on 13 September 2022).

## 4. iMSCs as a Source of EVs for Regenerative Medicine

### 4.1. iMSCs as a Source of EVs

The therapeutic interest of EVs from tissue-derived MSCs has been documented in many preclinical disease models and is being evaluated in clinics (for review, see [46]). iMSCs are emerging as a strong candidate for a new source of MSCs that might be suitable to replace adult MSCs. In addition to their therapeutic benefits, several features make iMSCs worth considering as a source of EVs, including the noninvasive way of generating iPSCs with a theoretically unlimited expansion rate, which could represent an advantage for the obtention of an inexhaustible source of EV preparations with homogeneous cargo.

The characteristics of EVs isolated from iMSCs (iMSC-EVs) resemble those of tissue-derived MSCs. Both types of EVs display similar features in terms of size (50–200 nm size range), production rates, ultrastructure, and proteome, including expression of TSG101, CD81, CD63, and CD9 [47,48,49]. Further functional analysis has also reported that iMSCs secrete EVs with biological functions that can affect target cells and ameliorate several aspects in disease models.

### 4.2. Therapeutic Efficacy of iMSC-EVs in Preclinical Disease Models

There is an increasing amount of preclinical data highlighting the application of iMSC-EVs, which are shown to have bioactive properties as good as—or sometimes even better than—the EVs from tissue-derived MSCs. The effects of iMSC-EVs were studied in a range of different disease models, from inflammatory and autoimmune diseases to tissue regeneration and cancer.

In the rat **tendinopathy** model, the local injection of iMSC-EVs was effective in alleviating pain. In vitro, iMSC-EVs significantly enhanced tenocyte proliferation and decreased the expression of proinflammatory markers, such as tumor necrosis factor-α (TNF-α), interleukin-1β (IL-1β), IL-6, and nerve growth factor (NGF), and in vivo, disease characteristics were improved both in pain and at the histological level [50]. Another study confirmed the therapeutic effect of iMSC-EVs on acute pain in rat tendinopathy by inhibiting mast cell activation through the regulation of the hypoxia-inducible factor-1α (HIF-1α) signaling pathway [51].

**Asthma and allergic rhinitis** are characterized by airway inflammation, and Group 2 innate lymphoid cells (ILC2s) are reported to play a critical role in these diseases. In vitro studies with patient-derived ILC2s revealed that small iMSC-EVs, but not EVs from fibroblasts, inhibited the function of activated ILC2s [52]. In the asthma mouse model, the number of total inflammatory cells, eosinophils, and neutrophils was significantly reduced following the administration of iMSC-EVs. Among the miRNAs that were present in iMSC-EVs, miR-146a-5p was significantly highly expressed in iMSCs compared with fibroblasts. EVs from iMSCs transfected with a miR-146a-5p inhibitor lost their effect on ILC2s, suggesting that the delivery of miR-146a-5p may be important for the role of iMSC-EVs on ILC2 suppression.

**Sjögren’s syndrome** (SS) is a chronic autoimmune disease affecting mainly salivary and lacrimal glands for which there is no effective treatment available. iMSC-EVs suppressed the immune cell activation in vitro, as well as the expression of proinflammatory factors that are relevant for SS progression [53]. Moreover, in the animal model, the infusion of iMSC-EVs decreased the lymphocyte infiltration in salivary glands, similar to the infusion of BM-MSCs, through the inhibition of antigen-presenting cells (APCs) and the downregulation of CD40 and inducible T cell costimulator ligand while upregulating IL-10 expression in damaged tissue. A subsequent study revealed that EVs from early passage iMSCs had better immunomodulatory potency than EVs from late-passage cells [54]. In vitro, iMSC-EVs from passage 5 were more effective than those from passage 15 in suppressing the secretion of inflammatory cytokines IFN-γ, IL-6, and IL-17, as well as increasing the secretion of transforming growth factor-β1 (TGF-β1). In the SS mouse model, they decreased with higher efficacy lymphocyte infiltration and the expression levels of the Th1 and Th17 markers. Moreover, the modulation of TGF-β1, miR-21, and miR-125b levels in iMSCs significantly affected the immunosuppressive effects of EVs in vivo.

In the experimental **autoimmune prostatitis** model, iMSC-EVs could significantly decrease inflammatory cell infiltration and promote basal lamina and glandular epithelial tissue repair [55]. Studies in mice showed reduced signs of chronic pelvic pain and improved void dysfunction. Mechanistically, the authors showed that the expression of cyclo-oxygenase-2 (COX-2) was downregulated with iMSC-EVs together with the decrease in Th1 and Th17 cells, while the percentage of Treg cells was increased.

In **skin regeneration**, iMSC-EVs and EVs from UC-MSCs were reported to exert a beneficial effect on proliferation, collagen secretion, and fibronectin expression in human keratinocytes and dermal fibroblasts [56]. iMSC-EVs were more efficient in increasing proliferation and fibronectin release in keratinocytes. Subcutaneous injections of iMSC-EVs around the wound sites in a rat model accelerated re-epithelialization, reduced scar widths, and promoted collagen deposition with increased blood vessel formation.

In different models of **vascular diseases**, iMSC-EVs have been associated with therapeutic potential. Intravenous administration of iMSC-EVs significantly attenuated aging-related arterial stiffness and hypertension, enhancing endothelium-dependent vascular relaxation and arterial compliance in old mice [57]. An analysis of kidney samples revealed that iMSC-EVs also diminished aging-related glomerulosclerosis, mesangial matrix expansion, increased collagen deposition, and glomerular collapse. In vitro data suggested that iMSC-EVs activated the SIRT1-AMPKα-eNOS pathway and inhibited the production of metalloproteinases (MMPs) and elastase. In a mouse model of limb ischemia, an intramuscular injection of iMSC-EVs markedly enhanced microvessel density and blood perfusion, consistent with the attenuation of ischemic injury [47]. In addition, iMSC-EVs activated angiogenesis-related molecule expression and promoted migration, the proliferation of human umbilical vein endothelial cells (HUVECs), and tube formation. Ischemia amelioration was also observed after iMSC-EV injection in an osteonecrosis mouse model [58]. Increased microvessel density in the femoral head led to reduced bone loss, and this effect was related to the activation of the PI3K/Akt signaling pathway in endothelial cells. The effects of iMSC-EVs on brain ischemia were also investigated in a rat model of stroke induced by middle cerebral artery occlusion [59]. iMSC-EVs significantly reduced brain tissue loss, enhanced angiogenesis, and improved neurological outcomes. In an in vitro ischemia model, iMSC-EVs rescued declined endothelial functions, such as migration and tube formation, and inhibited ischemic stroke-provoked autophagy via Stat-3-dependent pathway. Finally, amelioration of hepatic ischemia induced by reperfusion injury was reported after systematic administration of iMSC-EVs in mouse and rat models [60,61]. iMSC-EVs diminished hepatocyte necrosis and sinusoidal congestion, as well as the levels of liver injury markers aspartate aminotransferase (AST) and alanine aminotransferase (ALT) in mouse serum. A potential mechanism for iMSC-EVs is the activation of sphingosine kinase and the sphingosine-1-phosphate signaling pathway. Altogether, the findings suggest that iMSC-EVs promote angiogenesis under ischemic conditions.

**Corneal diseases** can cause corneal opacity. The association of iMSC-EVs with a thermosensitive chitosan-based hydrogel to extend the retention time and prevent the rapid drainage of vesicles in situ was tested [62]. A histological analysis of rat cornea after anterior lamellar damage showed that hydrogels loaded with iMSC-EVs promoted the repair of damaged corneal epithelium and the stromal layer, downregulating collagen expression in the corneal stroma and reducing scar formation. An analysis of miRNA expression suggested that the miR-432-5p contained in EV cargo can prevent extracellular matrix deposition on the side of injury. These findings can lay the foundations for the clinical use of EV-loaded thermosensitive hydrogels.

In **osteoarthritis**, iMSC-EVs attenuated the histological symptoms in cartilage from osteoarthritic mice and showed superior regenerative effects compared to synovium-derived MSCs [49]. Moreover, the in vitro analysis revealed that iMSC-EVs may have a stronger effect on chondrocyte proliferation and migration than tissue-derived MSCs.

**Osteoporosis** has a complex pathogenesis that comprises excessive bone resorption and deficient bone formation and vascularization, which are generally difficult to address with conventional therapies. Ovariectomized rats were used as a model for postmenopausal osteoporosis [63]. In vitro, iMSC-EVs enhanced proliferation and alkaline phosphatase activity of BM-MSCs from osteoporotic mice and upregulated the expression of osteoblast-related genes. In vivo experiments revealed that iMSC-EVs dramatically stimulated bone regeneration and angiogenesis in critical-sized calvaria defects of ovariectomized rats. In another study, Cui and collaborators developed an EV-delivery system based on engineered iMSC-EVs loaded with the siRNA of the Schnurri-3 (*Shn3)* gene and coated with bone-targeting peptides in the membrane to treat osteoporosis in ovariectomized mice [64]. The engineered iMSC-EVs could deliver the siRNA specifically to osteoblasts and mediated *Shn3* gene silencing, which led to enhanced osteogenic differentiation, neovascularization, and inhibition of osteoclast formation.

**Intervertebral disc****degeneration** models can be established by puncturing discs from the tails in rats. A rejuvenation potential of iMSC-EVs was shown after intervertebral disc injection, as shown by the attenuation of intervertebral disc degeneration and rejuvenation of senescent nucleus pulposus cells as assessed using magnetic resonance imaging (MRI) and histological analysis [65]. The antiaging effects were related to the delivery of miR-105-5p to nucleus pulposus cells and the activation of the sirtuin-6 (Sirt-6) pathway.

The fact that **critical bone defects** do not heal spontaneously has prompted the development of biomaterials to fill the defect. Tricalcium phosphate (β-TCP) is a synthetic and biodegradable ceramic material that has been shown to be especially effective when used in combination with osteoinductive substances (reviewed in [66]). The association of β-TCP with iMSC-EVs resulted in histologically proven bone regeneration in critical-sized calvaria bone defects in rats [67]. In vitro assays showed that iMSC-EVs were released from β-TCP scaffolds and internalized by target cells. These findings place the combination of iMSC-EVs and β-TCP scaffolds as a valid therapeutic prospect for clinical application in the bone defect.

Finally, bioinspired EV-mimicking nanoparticles can be obtained from intact cells as a different approach for exploring the beneficial effects of cell-derived vesicles. Using serial extrusion of iMSCs through filters with diminishing pore sizes, nanovesicles with EV characteristics can be recovered and loaded with agents, such as anticancer drugs [68]. In a model of **metastatic prostate cancer**, iMSC-EVs mimicking nanovesicles can be more selectively taken up by the prostate cancer cells and accumulate in the tumors as compared to iMSC-EVs or liposomes. Moreover, they can effectively deliver docetaxel, which is the standard drug for metastatic prostate cancer, to the target cells with reduced systemic cytotoxicity compared to free docetaxel, as demonstrated by a much higher systemic white blood cell count. These data introduce iMSC-EV mimetic nanovesicles as a promising platform for targeted delivery of anticancer agents.

## 5. Conclusions and Perspectives

Early phase clinical trials based on iPSCs or EVs are being conducted to demonstrate the safety and evaluate the efficacy of these approaches (for review, see [69]). Within the past decade, evidence has accumulated that EVs can display numerous biological properties that vary according to their parental cells. Although many questions remain regarding EV biology, initial clinical trials investigating EV therapeutics have begun, supported by internationally harmonized regulatory frameworks as discussed in the position paper of the International Society of Extracellular Vesicles [70]. The next step might be to evaluate the interest of iMSC-EVs in clinical trials. The publications cited in the previous section outline an extensive and growing list of potential applications of iMSC-EVs in the treatment of different diseases. Recent research has shown beneficial effects of iMSC-EVs per se, but also as engineered products when associated with the manipulation of iMSCs by genetic engineering or priming, post-release modifications, or combinations with three-dimensional scaffolds. The iMSC-EV research field has now much to unveil regarding the mechanisms of action and the applicability for clinical uses that still face some challenges.

iPSC-based therapy requires to constitute banks of allogeneic iPSCs that will allow unlimited access to pluripotent cells that can be differentiated into EV-producing MSCs. Such banks should match the HLA proteins between donors and recipients to prevent the stimulation of natural killer (NK) cells and reactive T lymphocytes or be used to generate a “universal” cell bank by genetic engineering. Different strategies have been developed to modulate the immunogenicity of donor iPSCs, including the overexpression of the HLA-E gene at the beta-2-microglobulin (BM2) locus in BM2 knockout iPSCs or the knockout of HLA-A and -B genes (retaining HLA-C gene expression), using CRISPR/Cas9 technology [71,72]. The risk of immune recognition of EVs is reported to be very low, but it could be further lowered by using genetically engineered iPSCs to generate iMSC-EVs.

There are many manufacturing requirements to be fulfilled for the clinical-grade production of EVs (for review, see [1,73]). Even though reaching high cell numbers can be more straightforward when considering iMSCs instead of tissue-derived MSCs, a large-scale EV production requires major MSC expansion. Closed system methods based on three-dimensional cell culture, such as bioreactors, can provide large amounts of clinical-grade iMSC-EVs. EV isolation methods must also be escalated to process large volumes of conditioned media but still ensure batch-to-batch reproducibility and lot consistency. An attractive method for EV purification is serial filtration based on the technology of tangential flow filtration that enables EV isolation under GMP-compliant conditions (for review, see [1]). It preserves the structural integrity of EVs and, depending on the molecular cut-off of the filters used, it allows to tailor the size of EVs and soluble proteins that are coisolated during the process. The huge advantage of EVs versus cell-based therapy is the possibility to optimize storage conditions by lyophilization, thereby minimizing the loss of active materials and preserving the functional activity of the product [74].

The safety of the drug recipient is the goal of early-stage clinical trials, and it requires extensive product testing to address possible undesired adverse effects. From the regulatory perspective, it is essential to have a strict characterization of the EV preparation content in terms of identity, purity, potency, and safety. The potential risks associated with EV manufacturing are the presence of coisolated residues, such as virus contaminations or endotoxins. Concerning iMSC-EVs, demonstration that reprogramming factors are absent both in the producing iMSCs and in the EV cargo would be mandatory. For assuring safety and efficacy, issues regarding the EV cargo, mechanism of action, pharmacokinetics, and biodistribution, as well as the administration and dosage will need to be addressed in relevant animal models, preferentially large animal models, before clinical translation of iMSC-EVs.

In conclusion, since their initial discovery 30 years ago, EVs derived from MSCs have gained great attention and are being evaluated in clinics for different applications involving tissue regeneration and immune modulation. The therapeutic interest of iMSC-EVs has been investigated in many preclinical models, and efficacy has been vigorously demonstrated by several research teams. To allow clinical translation, specific regulatory frameworks and several technical and biological improvements are still needed, but iMSC-EV-based approaches will certainly gain from the current trials evaluating MSC-EVs and iPSCs in many diseases.

## Figures and Tables

**Figure 1 biomedicines-10-02281-f001:**
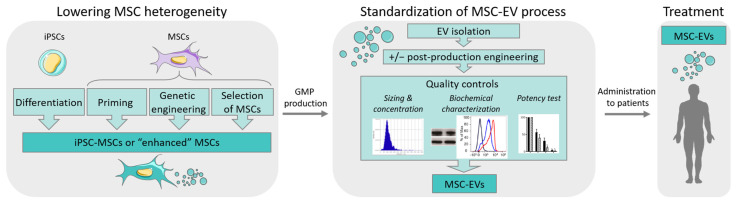
Strategies for optimizing MSC-EV production in regenerative medicine.

**Table 1 biomedicines-10-02281-t001:** Ongoing clinical trials using MSC-EVs.

Title	Source of EVs	Conditions	Status	Locations	NCT Number
iExosomes in Treating Participants with Metastatic Pancreas Cancer with KrasG12D Mutation	MSC-derived Exosomes with KRAS G12D siRNA	Pancreatic Adenocarcinoma	Recruiting	United States	NCT03608631
Allogenic Mesenchymal Stem Cell Derived Exosome in Patients with Acute Ischemic Stroke	MSC-derived exosomes enriched by miR-124	Cerebrovascular Disorders	Recruiting	Iran	NCT03384433
The Effect of Stem Cells and Stem Cell Exosomes on Visual Functions in Patients with Retinitis Pigmentosa	WJ-MSC	Retinitis Pigmentosa	Not yet recruiting	Turkey	NCT05413148
Safety and Efficacy of Injection of Human Placenta Mesenchymal Stem Cells Derived Exosomes for Treatment of Complex Anal Fistula	Placenta-MSC	Fistula Perianal	Recruiting	Iran	NCT05402748
A Pilot Clinical Study on Inhalation of Mesenchymal Stem Cells Exosomes Treating Severe Novel Coronavirus Pneumonia	ASC	Coronavirus	Completed	China	NCT04276987
A Clinical Study of Mesenchymal Stem Cell Exosomes Nebulizer for the Treatment of ARDS	MSC	Acute Respiratory Distress Syndrome	Recruiting	China	NCT04602104
A Clinical Study of Mesenchymal Progenitor Cell Exosomes Nebulizer for the Treatment of Pulmonary Infection	ASC	Drug-resistant	Recruiting	China	NCT04544215
A Tolerance Clinical Study on Aerosol Inhalation of Mesenchymal Stem Cells Exosomes in Healthy Volunteers	ASC	Healthy	Completed	China	NCT04313647
Exosome of Mesenchymal Stem Cells for Multiple Organ Dysfuntion Syndrome after Surgical Repair of Acute Type a Aortic Dissection	UC-MSC	Multiple Organ Failure	Not yet recruiting	China	NCT04356300
MSC-Exos Promote Healing of MHs	UC-MSC	Macular Holes	Active, not recruiting	China	NCT03437759
Clinical Efficacy of Exosome in Degenerative Meniscal Injury	SF-MSC	Knee Meniscus Injury	Recruiting	Turkey	NCT05261360
Effect of UMSCs Derived Exosomes on Dry Eye in Patients with cGVHD	UC-MSC	Dry Eye	Recruiting	China	NCT04213248
Efficacy and Safety of EXOSOME-MSC Therapy to Reduce Hyper-inflammation in Moderate COVID-19 Patients	MSC	SARS-CoV2 Infection	Recruiting	Indonesia	NCT05216562
Safety and Effectiveness of Placental Derived Exosomes and Umbilical Cord Mesenchymal Stem Cells in Moderate to Severe Acute Respiratory Distress Syndrome (ARDS) Associated with the COVID-19	UC-MSC	COVID-19/Acute Respiratory Distress Syndrome	Recruiting	United States	NCT05387278
Study of MSC-Exo on the Therapy for Intensively Ill Children	MSC	Sepsis/Critical Illness	Not yet recruiting	China	NCT04850469
The Use of Exosomes for the Treatment of Acute Respiratory Distress Syndrome or Novel Coronavirus Pneumonia Caused by COVID-19	MSC	COVID-19/Acute Respiratory Distress Syndrome	Not yet recruiting	United States	NCT04798716
Intra-articular Injection of MSC-derived Exosomes in Knee Osteoarthritis (ExoOA-1)	UC-MSC	Knee Osteoarthritis	Not yet recruiting	Chile	NCT05060107
the Safety and the Efficacy Evaluation of Allogenic Adipose MSC-Exos in Patients with Alzheimer’s Disease	ASC	Alzheimer’s Disease	Recruiting	China	NCT04388982
Effect of Microvesicles and Exosomes Therapy on Î²-cell Mass in Type I Diabetes Mellitus (T1DM)	UC-MSC	Diabetes Mellitus Type 1	Unknown status	Egypt	NCT02138331
MSC EVs in Dystrophic Epidermolysis Bullosa	BM-MSC	Dystrophic Epidermolysis Bullosa	Not yet recruiting	Aegle Therapeutics	NCT04173650
Safety and Efficiency of Method of Exosome Inhalation in COVID-19 Associated Pneumonia	MSC	COVID-19/SARS-CoV-2 Pneumonia	Enrolling by invitation	Russian Federation	NCT04602442
Evaluation of Safety and Efficiency of Method of Exosome Inhalation in SARS-CoV-2 Associated Pneumonia.	MSC	COVID-19/SARS-CoV-2 Pneumonia	Completed	Russian Federation	NCT04491240
Extracellular Vesicle Infusion Treatment for COVID-19 Associated ARDS	BM-MSC	COVID-19/Acute Respiratory Distress Syndrome/Pneumonia	Completed	United States	NCT04493242
Bone Marrow Mesenchymal Stem Cell Derived Extracellular Vesicles Infusion Treatment for ARDS	BM-MSC	Acute Respiratory Distress Syndrome	Not yet recruiting	Direct Biologics	NCT05127122
A Global Expanded Access Protocol on Bone Marrow Mesenchymal Stem Cell Derived Extracellular Vesicle Infusion Treatment for Patients with COVID-19 Associated ARDS	BM-MSC	COVID-19/Acute Respiratory Distress Syndrome	Available	Direct Biologics	NCT04657458
Safety of Mesenchymal Stem Cell Extracellular Vesicles (BM-MSC-EVs) for the Treatment of Burn Wounds	BM-MSC	Burns	Not yet recruiting	Aegle Therapeutics	NCT05078385
A Phase I Study of ExoFlo, an ex Vivo Culture-expanded Adult Allogeneic Bone Marrow Mesenchymal Stem Cell Derived Extracellular Vesicle Isolate Product, for the Treatment of Medically Refractory Crohn’s Disease	BM-MSC	Crohn’s Disease/Irritable Bowel Disease	Not yet recruiting	Direct Biologics	NCT05130983
Bone Marrow Mesenchymal Stem Cell Derived Extracellular Vesicles Infusion Treatment for Mild-to-Moderate COVID-19: A Phase II Clinical Trial	BM-MSC	COVID-19	Not yet recruiting	Austin, United States	NCT05125562
ExoFlo^TM^ Infusion for Post-Acute COVID-19 and Chronic Post-COVID-19 Syndrome	BM-MSC	COVID-19/Postviral Syndrome/Dyspnea	Not yet recruiting	Direct Biologics	NCT05116761
Bone Marrow Mesenchymal Stem Cell Derived Extracellular Vesicles as Early Goal Directed Therapy for COVID-19 Moderate-to-Severe Acute Respiratory Distress Syndrome (ARDS): A Phase III Clinical Trial	BM-MSC	COVID-19/Acute Respiratory Distress Syndrome	Not yet recruiting	Direct Biologics	NCT05354141
A Safety Study of IV Stem Cell-derived Extracellular Vesicles (UNEX-42) in Preterm Neonates at High Risk for BPD	BM-MSC	Bronchopulmonary Dysplasia	Terminated	United States	NCT03857841
Study of ExoFlo for the Treatment of Medically Refractory Ulcerative Colitis	BM-MSC	Ulcerative Colitis	Not yet recruiting	Direct Biologics	NCT05176366
Intermediate Size Expanded Access for the Use of ExoFlo in the Treatment of Abdominal Solid Organ Transplant Patients Who Are at Risk of Worsening Allograft Function with Conventional Immunosuppressive Therapy Alone	BM-MSC	Solid Organ Transplant Rejection	Not yet recruiting	Direct Biologics	NCT05215288

MSC: MSC from unknown source; BM-MSC: Bone Marrow-derived MSC; ASC-MSC: Adipose Tissue-derived MSC; UC-MSC: Umbilical Cord-derived MSC; SF-MSC: Synovial Fluid-derived MSC; WJ-MSC: Wharton Jelly-derived MSC.

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
