# Peer review of "Recent Advances in Extracellular Vesicle-Based Therapies Using Induced Pluripotent Stem Cell-Derived Mesenchymal Stromal Cells"

_biomedicines, 2022, doi:10.3390/biomedicines10092281_

Round 1

Reviewer 1 Report

The authors present in this review the interesting proposal that EVs derived from homogenous populations of MSCs derived from iPSCs. Although the proposal is logic and not new, the review has its merit as the authors approach main issues of reduced rejection when compared to MSC-based therapeutics, similar senescence rates as compared to MSCs isolated from donors supporting safety of EV-based therapeutics and encouraging preclinical data for these types of treatments. The information gathered is complete, well-structured and easy to read. The aspects of bioreactors for massive production or banking of iPSCs to ensure an unlimited resource of iMSCs and therefore of iMSC-produced EVs appear covered. Nevertheless, since a key aspect of the review is to obtain MSCs of homogenous quality from iPSCs. It is recommended the authors incorporate the following review on methods:

Dupuis V, Oltra E. Methods to produce induced pluripotent stem cell-derived mesenchymal stem cells: Mesenchymal stem cells from induced pluripotent stem cells. World J Stem Cells. 2021 Aug 26;13(8):1094-1111. doi: 10.4252/wjsc.v13.i8.1094. PMID: 34567428; PMCID: PMC8422924.

Attention must be paid for the limitations to obtain MSCs of homogenous quality with the desired phenotypic features according to the target treatment, as commented in the referred paper.

Minor details:

Please review Table 1 format (Cont. on first appearance?) NCT appears in two lines

Unify nomenclature for EVs, not “Evs”

Perhaps is convenient adopting the nomenclature iMSC to indicate MSCs derived from iPSC

Author Response

We thank the reviewer for this recommendation. The referenced paper is now discussed in the main text (see page 7) and has been added in the list of references.

Minor details:

Please review Table 1 format (Cont. on first appearance?) NCT appears in two lines

We apologize for the column with the confusing numeration that should not be part of the table and is now deleted in the new version.

Unify nomenclature for EVs, not “Evs”

The typing error has been corrected.

Perhaps is convenient adopting the nomenclature iMSC to indicate MSCs derived from iPSC

 Thank you again for the recommendation. It will help the understanding of the manuscript.

Reviewer 2 Report

This manuscript is comprehensive review on EVs derived form iPSC-MSC, especially, therapeutic and functional role of extracellular vesicles (EVs) for diverse disease as well as stemness. This manuscript has systematically reviewed the recent literature which is up-to-date,and provided an exhaustive and informative summary on this issue. In particular, the authors pointed the usefulness, feasibility and advantges of iMSC-EVs over conventional MSCs. In my opinion, the manuscript deserves publication in the journal Biomedicines.

- Minor comments

(1) In the section 2, definition or biogenesis of microvesicles or microparticles needs to be rephrased. Microvesicles are known to be produced by outward budding of plasma membrane.

(2) In many cases, the examples of the quotation were bluntly expressed (i.e., (dendritic cells (DC), tumor cells, …). These should be expressed as (dendritic cells (DC), tumor cells, etc)

Author Response

- Minor comments

(1) In the section 2, definition or biogenesis of microvesicles or microparticles needs to be rephrased. Microvesicles are known to be produced by outward budding of plasma membrane.

We thank the reviewer for his critical reading and the notification of this error. Of course, microvesicles are not release upon fusion with the plasma membrane. The sentence has been corrected (see page 4).

(2) In many cases, the examples of the quotation were bluntly expressed (i.e., (dendritic cells (DC), tumor cells, …). These should be expressed as (dendritic cells (DC), tumor cells, etc)

              The sentences have been changed accordingly.